# FedSDR: Federated Self-Distillation with Rectification

**Ziheng Ren** [* 1]  **Zhanming Shen** [* 2]  **Hao Wang** [3]  **Ning Liu** [† 4]  **You Song** [† 1]

## Abstract

Federated fine-tuning of Large Language Models faces severe statistical heterogeneity. However, existing model-level defenses often overlook the root cause: intrinsic data distribution mismatches. In this work, we first establish **Federated Self-Distillation (FedSD)** as a fundamental and potent strategy. By projecting client representations into a smoothed "model-understanding space," FedSD alone serves as a universal booster, demonstrating superior performance over conventional algorithms. Despite its success, we identify a subtle trade-off termed the **Rewrite Paradox**—unconstrained self-distillation can inadvertently increase hallucinations and redundancy. To refine this paradigm, we further propose **FedSDR** (Federated Self-Distillation with Rectification), the ultimate reinforced framework. It augments FedSD with a dual-stream mechanism: a local **LoRA-S (Smoothing)** branch to *implicitly* absorb heterogeneity via distilled data, and a parallel global **LoRA-R (Rectification)** branch anchored to raw data to enforce factual correctness. By selectively aggregating only LoRA-R, FedSDR yields a globally aligned and faithful model. Extensive experiments verify its superior performance.

## 1. Introduction

Federated Learning (FL) has emerged as a paramount paradigm for fine-tuning Large Language Models (LLMs) (McMahan et al., 2017) by leveraging decentralized data while adhering to strict privacy regulations (Yao et al., 2024). By collaboratively aggregating updates from isolated clients, FL enables the cultivation of global intelligence without exposing private raw data (Ren et al., 2024; Charles et al., 2024). However, the efficacy of FL is severely compro-

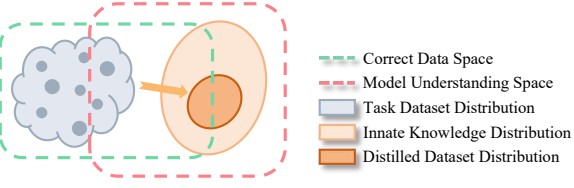

*(a)* Geometric Interpretation of Data Refinery

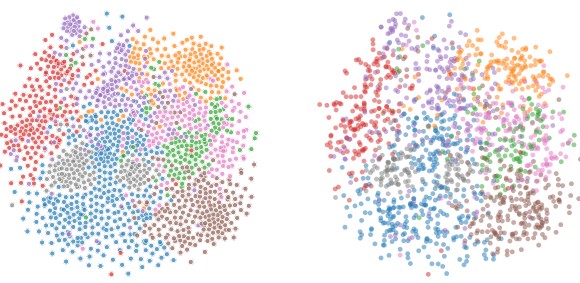

*(b)* Raw Data: Heterogeneous   *(c)* Distilled: Aligned

*Figure 1.* **The Data Refinery Process.** (a) **Conceptual Illustration**: FedSD leverages the model's *Innate Knowledge Distribution* as a universal manifold to re-map disjoint, client-specific raw data (gray) into a unified *Model Understanding Space* (red), transforming sharp data boundaries into a smoothed manifold. (b-c) **Empirical Validation**: t-SNE visualization on the Databricks Dolly-15K dataset confirms the "manifold flattening" effect, where self-distillation dissolves distinct client clusters into a shared, synchronized distribution to facilitate stable global aggregation.

mised by **statistical heterogeneity** (Zhao et al., 2018; Li et al., 2020). In real-world scenarios, the data distributions across clients vary drastically, leading to divergent optimization trajectories ("client drift") and catastrophic degradation of the global model performance.

To mitigate this, the community has developed a plethora of Personalized Federated Learning (PFL) strategies. Classical defenses (e.g., FedProx(Li et al., 2020), SCAFFOLD(Karimireddy et al., 2020)) constrain local updates to reduce client drift. By contrast, more recent approaches like FedDPA (Yang et al., 2024a) use dual LoRA to separate a global adapter from client-specific personalization (Wu et al., 2024b; Yi et al., 2025; Shen et al., 2025c; Seo et al.). However, we argue that they still focus on a **model-centric** perspective—treating the symptoms (weight diver-

*Equal contribution [1]Beijing University of Aeronautics and Astronautics [2]Zhejiang University [3]Stevens Institute of Technology [4]Shandong University. Correspondence to: Ning Liu <liun21cs@sdu.edu.cn>, You Song <songyou@buaa.edu.cn>.

*Proceedings of the 43rd International Conference on Machine Learning*, Seoul, South Korea. PMLR 306, 2026. Copyright 2026 by the author(s).

gence) rather than the root cause: the **intrinsic distribution mismatch** of the data itself. A fundamental question thus remains largely unexplored (Qin et al., 2025): *Is it possible to align the heterogeneous data distributions at the source level, rather than passively correcting the model after divergence has occurred?*

Recent advances in Self-Distillation offer a compelling answer (Wang et al., 2023; Yang et al., 2024b). We find that the model itself may act as the most effective "rectifier" for training data. As conceptually and empirically illustrated in **Figure 1**, we propose a data-centric paradigm shift. We hypothesize that an LLM can leverage its *Innate Knowledge Distribution* as a universal manifold to re-map disjoint, client-specific raw data. By projecting heterogeneous distributions into a unified *Model Understanding Space*, our proposed **FedSD** effectively transforms sharp, non-overlapping data boundaries into a smoothed, distilled manifold. Our t-SNE analysis (Figure 1a-b) confirms this "manifold flattening" effect: **self-distillation dissolves the distinct clusters of raw client data, pulling diverse local manifolds into a shared distribution.** This alignment facilitates a synchronized optimization landscape, fundamentally mitigating the divergence caused by statistical heterogeneity and making the global aggregation far more conducive (Table 1 & 2).

While FedSD achieves remarkable alignment, a deeper investigation reveals a subtle side-effect inherent to unconstrained distillation, which we term the **Rewrite Paradox**. We observe that while training on self-distilled data smooths the optimization landscape, it sometimes also encourages the model to generate hallucinations and reinforces stylistic redundancies (Gudibande et al., 2023). The "smoothed" data, while easier to learn, may lose the sharp precision of the ground truth.

To resolve this dilemma and evolve FedSD into a **more comprehensive** architecture, we further propose **FedSDR (Federated Self-Distillation with Rectification)**. The core intuition is to separate "alignment" from "fidelity." We introduce a novel Dual-Stream mechanism, i.e., LoRA-S and LoRA-R. LoRA is used as a smoothing adapter to implicitly absorb the distributional heterogeneity and the noise within the distilled stream while LoRA-R is used as a parallel recification adapter to strictly focus on learning factual correctness from raw data. By selectively aggregating only the rectified parameters, FedSDR effectively filters out the noise, retaining only the refined consensus.

We present, to our knowledge, the first systematic study of LLM personalization in FL from a data-centric perspective, revealing intrinsic client data distribution mismatch as **a primary bottleneck** in PFL:

**1.** From a data-centric viewpoint, we identify the model itself as the most effective rectifier of training data, and

show that **FedSD (Federated Self-Distillation)** yields substantial gains when directly applied across all classical FL algorithms.

**2.** We *first* uncover the "Rewrite Paradox," showing that naive self-distillation may face factual hallucinations and stylistic verbosity, and we further propose **FedSDR** to decouple distribution smoothing from factual rectification.

**3.** Extensive experiments show that FedSDR achieves state-of-the-art performance across various Non-IID regimes; moreover, we find that implicit smoothing from self-distilled data in LoRA-S substantially eases raw-data learning in LoRA-R, further highlighting self-distillation as a promising new paradigm for personalized federated learning.

## 2. Related work

### 2.1. Federated Fine-Tuning of LLMs and Personalization

The intersection of Federated Learning and Large Language Models has attracted significant attention as a privacy-preserving paradigm for instruction tuning (Borazjani et al., 2025). Given the prohibitive computation and communication cost of full-parameter fine-tuning, Parameter-Efficient Fine-Tuning (PEFT) techniques (Ye et al., 2024; Wu et al., 2024a; Qin et al., 2024), particularly LoRA (Hu et al., 2022; Shen et al., 2025c; Li et al., 2024), have become the standard. However, Mitigating the impact of statistical heterogeneity is a central theme in FL research. Prior personalized FL methods (Arivazhagan et al., 2019; Yi et al., 2023; Zhang et al., 2023) either regularize local updates or decouple shared and private components for personalization; more recent approaches such as FedDPA further adopt a dual-LoRA (Yang et al., 2024a; Jiaxing et al., 2024) design to separate a global adapter from client-specific personalization. **Crucially, both paradigms operate at the model level.** They only attempt to force alignment or allow divergence via parameter constraints. In contrast, our work proposes a **Data-Centric Paradigm Shift**. Instead of passively correcting model weights, we actively align the underlying data distributions via self-distillation, treating the root cause of heterogeneity directly. Taxonomic details and baseline comparisons are provided in Appendix A (Jiang & Lin, 2023; Tan et al., 2024; Bao et al., 2024; Li et al.).

### 2.2. Self-Distillation and Instruction Tuning

Knowledge Distillation (KD) (Lin et al., 2020; Ma et al., 2022; Shen et al., 2026; 2025b) has been explored in FL (e.g., FedGen , FedDistill) (Venkateswaran et al., 2023; Song et al., 2024) to transfer knowledge via logits, reducing communication costs. However, these methods often require public proxy datasets or complex generator training, which is impractical for LLMs. On the other hand, **Self-Distillation** in the context of LLMs (e.g., WizardLM (Xu

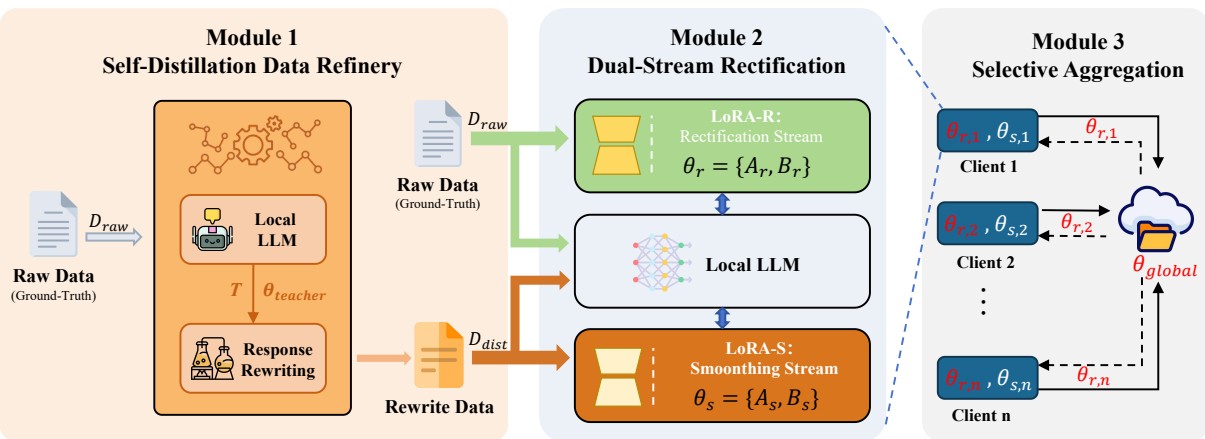

*Figure 2.* **The FedSDR Framework.** Our paradigm mitigates statistical heterogeneity through a three-stage refinery and rectification pipeline. **Module 1 (Data Refinery):** As shown in Figure 1a, heterogeneous correct raw data is projected into a unified "Model-Understanding Space" via self-distillation to generate aligned rewritten data. **Module 2 (Dual-Stream Rectification):** We adopt an *alternating optimization* strategy where **LoRA-S** first absorbs distributional noise and stylistic biases as a functional scaffold, while **LoRA-R** is subsequently trained on raw data to ensure factual correctness. **Module 3 (Selective Aggregation):** To avoid the "Rewrite Paradox," clients only upload the Rectification Adapter ($\Theta_r$). Since stylistic noise and hallucinations are absorbed locally by **LoRA-S**, this filtering ensures a globally aligned yet faithful consensus.

et al., 2023), Alpaca)-where a model refines its own training data (Wang et al., 2023; Yang et al., 2024b; Shen et al., 2025a)-has shown remarkable success in centralized settings for improving instruction quality (Wang et al., 2023). However, while prior works have primarily demonstrated that mapping data into a model-understanding space via self-distillation can effectively mitigate catastrophic forgetting (Yang et al., 2024b), our work further explores this mapping relationship and extends it to alleviate **statistical heterogeneity** in federated environments. We empirically prove that this alignment at the data level facilitates a more synchronized optimization landscape. Beyond its benefits, we also identify that naive self-distillation triggers a **"Rewrite Paradox"**—unconstrained rewriting inadvertently induces factual hallucinations and stylistic verbosity. To resolve this, we propose **FedSDR**, a dual-stream mechanism that decouples distribution alignment from factual fidelity.

## 3. Preliminaries and Motivation

In this section, we provide the foundational concepts of Federated Learning (FL) and Low-Rank Adaptation (LoRA), with formal formulations detailed in Appendix B. Building upon these preliminaries, we revisit the mechanism of self-distillation to empirically motivate our proposed dual-stream framework.

### 3.1. Background and Notations

We consider a standard federated setting with $K$ clients, where each client $k$ holds a private dataset $D_k$ drawn from a potentially Non-IID distribution $\mathcal{P}_k$ (Li et al., 2019). To

address the prohibitive costs of fine-tuning Large Language Models (LLMs) in such decentralized environments, we employ Low-Rank Adaptation (LoRA) (Hu et al., 2022) as our primary parameter-efficient tuning strategy. By freezing the pre-trained weights and only updating low-rank decomposition matrices, clients significantly reduce communication bandwidth. A rigorous formulation of the FL objective and the LoRA forward pass is provided in **Appendix B**.

### 3.2. Revisiting Self-Distillation: Alignment vs. Fidelity

FedSD has emerged as a promising technique to mitigate data heterogeneity. By utilizing the local model itself to generate soft labels (responses), FedSD projects diverse raw data into a model-understanding space. However, our empirical analysis reveals that this process is a double-edged sword.

**The Power: Reducing Heterogeneity via Data Alignment.** The core challenge in FL is the divergence of local optimization landscapes. As visualized in Figure1a-b, raw data from different clients forms disjoint clusters (Figure1b). Training directly on this leads to weight divergence. In contrast, Figure 1c demonstrates that self-distillation significantly smoothes the data distribution. By converting hard labels into soft, model-generated distributions, FedSD implicitly aligns the diverse local manifolds, making the optimization landscape far more conducive to aggregation. This confirms FedSD as a powerful, data-centric booster for FL.

To move beyond visualization-level observations, we further verify the alignment effect of self-distillation from both the *text-distribution* and *optimization* perspectives, using five

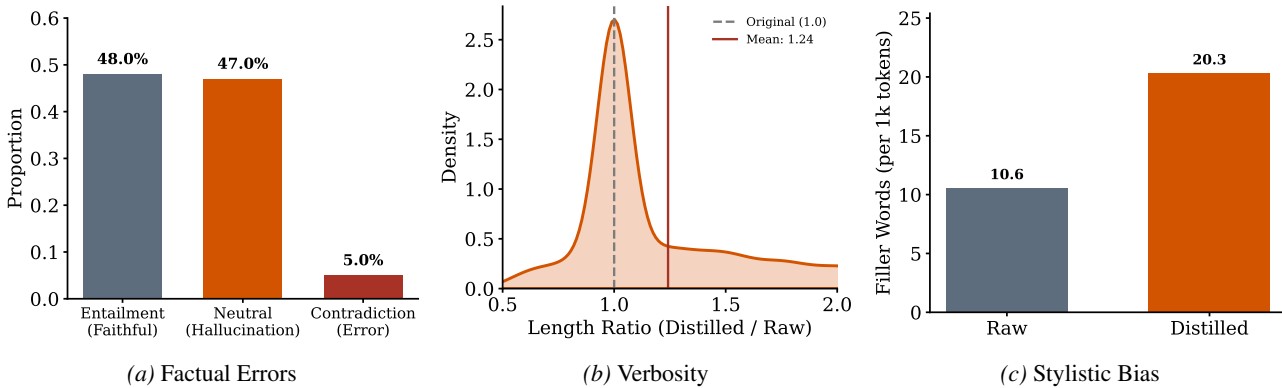

*Figure 3.* **The Rewrite Paradox.** (a) Distilled data introduces factual hallucinations. (b) Re-written responses become significantly longer, leading to redundancy. (c) The model reinforces its own stylistic biases, doubling the frequency of filler words.

highly heterogeneous subsets (FinGPT(Liu et al., 2023), MathInstruct(Yue et al., 2024), Code-Alpaca(Chaudhary, 2023), Alpaca(Taori et al., 2023), and MedAlpaca(Han et al., 2023)).

**Text-Distribution Evidence.** We measure inter-client divergence before and after self-distillation using Jensen–Shannon (JS) divergence and TF-IDF cosine similarity. As shown in Table 1, rewriting consistently reduces token-level distributional discrepancy and increases semantic overlap across clients, confirming that self-distillation compresses originally dispersed client expressions into a more unified representational space through the shared generative preference of the same teacher model.

*Table 1.* Inter-client text-distribution divergence before and after self-distillation.

| Metric | Raw | Distilled | $\Delta$ |
|---|---|---|---|
| JS Divergence ↓ | 0.4074 | 0.3611 | −0.0463 |
| TF-IDF Cosine Sim. ↑ | 0.6362 | 0.7064 | +0.0702 |

**Optimization-Level Evidence.** To further examine whether the alignment translates into more cooperative optimization, we analyze *cross-task gradient cosine similarity* and *cross-task loss transfer* across the same five subsets. For each source task, we compute the average off-diagonal change over the other four target tasks. Here, **Grad. Sim.** denotes the average change in cross-task gradient cosine similarity (Distilled − Raw, in percentage points; higher is better), and **Loss Trans.** denotes the average change in cross-task loss transfer (Distilled − Raw, in percentage points; more negative is better).

As shown in Table 2, all five source tasks exhibit positive average off-diagonal gains in gradient similarity, indicating that after distillation, update directions across tasks become more aligned overall. Meanwhile, all five tasks also show more negative average off-diagonal changes in loss transfer,

*Table 2.* Cross-task optimization alignment after self-distillation. Each row reports the average off-diagonal change over the other four target tasks.

| Source Task | Avg. Grad. Sim. ↑ | Avg. Loss Trans. ↓ |
|---|---|---|
| FinGPT | +41.5 pp | −2.5 pp |
| MathInstruct | +13.8 pp | −4.0 pp |
| Code-Alpaca | +13.6 pp | −4.9 pp |
| Alpaca | +10.8 pp | −5.1 pp |
| MedAlpaca | +8.1 pp | −5.4 pp |

meaning that training on one task more consistently helps reduce loss on other tasks rather than interfering with them. Together with the text-distribution evidence in Table 1, these analyses provide mechanism-level support for our interpretation: the benefit of self-distillation is not merely generic regularization, but that it **reshapes heterogeneous supervision into a more aligned and cooperative optimization space**, thereby alleviating cross-task interference under heterogeneous federated data distributions.

**The Problem: The Rewrite Paradox.** Despite its alignment benefits, relying solely on self-distilled data introduces semantic corruption, which we term the *Rewrite Paradox*. Crucially, we observe that this paradox is significantly more detrimental in the Federated setting than in centralized training. In FL, the subtle hallucinations and stylistic biases generated by individual clients are not merely local noises; they are aggregated and broadcasted by the server, potentially forming a *pseudo-consensus*. This "hallucination-aggregation" loop gradually dilutes the global model's factual precision over communication rounds, leading to a model that is "fluent but unfaithful." Representative case studies documenting these failure modes, are detailed in Appendix F. Specifically, our analysis reveals:

**(1) Factual Hallucinations:** As shown in Figure 3a, the model often rewrites correct facts into plausible but incor-

rect statements. Approximately 47% of the distilled samples fail to strictly entail the ground truth, polluting the training set with hallucinations.

**(2) Verbosity & Information Dilution:** Contrary to summarization, Figure 3b reveals that the model tends to generate responses that are longer on average. This verbosity often dilutes the core information density, making the optimization objective drift away from precision.

**(3) Inductive Bias Amplification:** The model acts as an echo chamber, reinforcing its own stylistic flaws. As seen in Figure 3c, the frequency of non-informative filler words doubles in the distilled data. This confirms that self-distillation injects the model's inherent inductive bias into the data.

**The Solution: Why Dual-LoRA and Selective Aggregation?** To reconcile the conflict between "distribution alignment" and "factual correctness," we propose a Dual-LoRA architecture with a selective aggregation strategy:

**(1) The Necessity of Dual-Streams:** Since distilled data carries the model's inductive biases (verbosity, hallucinations), training a single model on it would compromise factual capability. Thus, we introduce **LoRA-S** to learn from distilled data and **LoRA-R** to learn from raw data.

**(2) Rationale for Selective Aggregation:** We explicitly treat **LoRA-S** as a local "shock absorber." Its role is to absorb the heterogeneity and the model-specific stylistic noises (e.g., verbosity) revealed in our analysis. Since LoRA-S contains these defects, it must remain local. In contrast, **LoRA-R**, which benefits from the smoothed optimization landscape provided by LoRA-S but is strictly anchored to the ground truth, captures the clean, factually correct consensus. Therefore, by uploading only LoRA-R, we ensure the global model is both aligned and faithful.

## 4. Methodology

### 4.1. Overview: A Data-Centric Paradigm Shift

As illustrated in Figure 2, FedSDR operates on a "Generate-Align-Rectify" philosophy, structurally decoupled into three distinct modules:

**Module 1: Self-Distillation Data Refinery.** Instead of forcing the model to fit disjoint raw data directly, we first project the heterogeneous raw data into a smoothed "model-understanding space" via self-distillation. This process generates a standardized dataset that aligns the optimization landscapes across clients.

**Module 2: Dual-Stream Rectification.** To counteract the "Rewrite Paradox" (i.e., hallucinations and inductive biases) introduced by the distillation process, we design a dual-stream architecture. We assign a *Smoothing Stream* (LoRA-S) to absorb the distributional heterogeneity and a *Rectification Stream* (LoRA-R) to strictly anchor the model to factual ground truth.

**Module 3: Selective Aggregation.** We introduce a filtering mechanism during the server aggregation phase. By uploading only the rectified parameters (LoRA-R) and discarding the smoothing parameters (LoRA-S), we ensure the global consensus absorbs valid knowledge without inheriting the stylistic noise or hallucinations generated during local smoothing.

### 4.2. Module 1: Self-Distilled Data Refinery

The first module functions as a local pre-processing engine, transforming the raw, heterogeneous input space into a dual-view format that facilitates both alignment and fidelity.

**Teacher Construction & Distribution Probing** At the beginning of each communication round $t$, client $k$ receives the global model parameters $\Theta_{global}^t$. To maximize resource efficiency without relying on external large-scale models, we adopt a **Self-Distillation** strategy. The local model at the initial step of the round serves as its own "Teacher":

$$\Theta_{teacher} \leftarrow \Theta_k^{(t,0)} = \Theta_{global}^t. \tag{1}$$

For each sample $(x, y_{true})$ in the local raw dataset $D_{raw}$, the teacher model performs a forward inference pass.

**Federated Self-Distilled Fine-Tuning** In contrast to classical logit-based KD in FL, which often relies on public proxy data or additional generators, we adopt **self-distillation** tailored for instruction tuning. As shown in Eq. 2, each client prompts the seed LM to rewrite the original response $y_i$ into a distilled response $\tilde{y}_i$, thereby mapping the raw supervision into the seed LM's own response distribution. **Importantly, this rewriting is performed only once per client before federated training begins**, and the resulting distilled dataset is reused across all communication rounds.

In a formal sense, for client $k$ with local dataset $D_k = \{(c_i, x_i, y_i)\}_{i=1}^{n_k}$, we construct a distilled dataset:

$$D_k^{dist} = \{(c_i, x_i, \tilde{y}_i)\}_{i=1}^{n_k}, \quad \tilde{y}_i \sim f_{\Theta_{teacher}}(\cdot \mid c_i, x_i, y_i), \tag{2}$$

where $\Theta_{teacher}$ is the local teacher initialized as in Eq. 1.

**FedSD: Federated Self-Distilled Fine-Tuning** With the distilled dataset $D_k^{dist}$ constructed in Eq. 2, FedSD performs standard federated optimization using $\tilde{y}$ as supervision. At each communication round $t$, the server broadcasts the current global model $\Theta_{global}^t$ to participating clients, each of which performs local fine-tuning on $D_k^{dist}$ and returns updated parameters for aggregation.

*Local objective* Compared to standard local objective of federated learnig, we replace the original supervision with the distilled response. The local fine-tuning objective on

client $k$ is defined as Eq 3.

$$\mathcal{L}_{\text{FedSD}}^k(\Theta) = \frac{1}{n_k} \sum_{i=1}^{n_k} - \log f_\Theta(\tilde{y}_i \mid c_i, x_i). \quad (3)$$

*Federated optimization* At communication round $t$, each client performs local updates (e.g., $E$ steps of SGD) starting from the received global model:

$$\Theta_k^{t+1} \leftarrow \text{LocalUpdate}(\Theta_{global}^t, D_k^{dist}), \quad (4)$$

and the server aggregates client models (or PEFT parameters, if only adapters are trained) via weighted averaging:

$$\Theta_{global}^{t+1} \leftarrow \sum_{k=1}^{K} \frac{n_k}{\sum_{j=1}^{K} n_j} \Theta_k^{t+1}. \quad (5)$$

### 4.3. Module 2: Dual-Stream Rectification Architecture

While **FedSD** provides a powerful alignment engine that significantly boosts performance, we aim to further eliminate the subtle side effects identified in the "Rewrite Paradox" to achieve optimal fidelity. We propose a Dual-Stream Architecture that structurally decouples *Optimization Smoothing* from *Factual Rectification*.

**Decouple of Factual Rectification** We maintain the pretrained LLM backbone $W_0 \in \mathbb{R}^{d \times d}$ frozen to preserve general capabilities. To enable dual-stream learning, we inject two parallel pairs of Low-Rank Adaptation (LoRA) modules into the transformer attention layers. Let $h$ denote the input to a linear layer. The output $h_{out}$ is computed via a joint forward propagation:

$$h_{out} = W_0 h + \underbrace{\frac{\alpha}{r} B_r A_r h}_{\text{Stream R (Rectification)}} + \underbrace{\frac{\alpha}{r} B_s A_s h}_{\text{Stream S (Smoothing)}}, \quad (6)$$

where $\{A_r, B_r\}$ and $\{A_s, B_s\}$ are rank-$r$ trainable matrices. Crucially, these two streams operate as additive bypass connections in the activation space (Zhang et al., 2024).

**Alternating Dual-Stream Training Strategy** Although Stream R and Stream S are jointly injected and participate in the same forward pass (Eq. 6), they are **not** optimized by a simultaneous multi-objective. Instead, we adopt an **alternating (block-coordinate) training** scheme that explicitly decouples their roles.

Let $\Theta_r = \{A_r, B_r\}$ and $\Theta_s = \{A_s, B_s\}$ denote the trainable parameters of LoRA-R and LoRA-S, respectively. Denote by $p(\cdot \mid c, x; \Theta_r, \Theta_s)$ the next-token distribution induced by the backbone with the dual-stream LoRA injection.

*Stage 1: Train the Smoothing Stream ($\Theta_s$) on distilled data.*
We first update **LoRA-S** while **freezing LoRA-R**($\Theta_r$). This

stage uses the distilled dataset $D_{dist}$ with standard cross-entropy, encouraging $\Theta_s$ to absorb client-specific heterogeneity and stylistic biases in an *implicitly smoothed* optimization landscape:

$$\min_{\Theta_s} \mathcal{L}_{smooth}(\Theta_s \mid \Theta_r) = \mathbb{E}_{D_{dist}} \Big[ - \log p_{\Theta_r, \Theta_s}(\tilde{y} \mid c, x) \Big]. \quad (7)$$

*Stage 2: Train the Rectification Stream ($\Theta_r$) on raw data.*
After $\Theta_s$ is trained, we **freeze LoRA-S**($\Theta_s$) and update **LoRA-R** using the raw dataset $D_{raw}$ with cross-entropy. Benefiting from the stabilized representations shaped by Stage 1, $\Theta_r$ focuses on anchoring the model to factual ground-truth supervision:

$$\min_{\Theta_r} \mathcal{L}_{rect}(\Theta_r \mid \Theta_s) = \mathbb{E}_{D_{raw}} \Big[ - \log p_{\Theta_r, \Theta_s}(y \mid c, x) \Big]. \quad (8)$$

*Summary.* The local update on each client thus follows an alternating procedure:

$$\begin{aligned} \Theta_s &\leftarrow \arg\min_{\Theta_s} \mathcal{L}_{smooth}(\Theta_s \mid \Theta_r), \\ \Theta_r &\leftarrow \arg\min_{\Theta_r} \mathcal{L}_{rect}(\Theta_r \mid \Theta_s), \end{aligned} \quad (9)$$

where both stages share the same joint forward propagation, but only one stream is trainable at a time. This design makes $\Theta_s$ serve as an **implicit smoother** that reduces the learning difficulty of $\Theta_r$ on raw data, while preventing the distilled-induced biases from directly dominating the rectification stream.

### 4.4. Module 3: Selective Aggregation Strategy

This module revisits the global aggregation phase. As discussed in the "Rewrite Paradox," self-distilled supervision may carry stylistic noise (e.g., verbosity) and occasional hallucinations. In our design, such effects are **already absorbed locally** by the Smoothing Stream $\Theta_s$, which provides implicit smoothing during client-side optimization via the coupled forward pass. Therefore, to ensure that these stylistic biases do not interfere with globally correct learning, we adopt a **Selective Aggregation** protocol: clients only upload the Rectification Adapter. During the upload phase, clients separate the two streams:

**Local-only ($\Theta_s$):** The Smoothing Adapter is kept on-device (or re-initialized). It is used solely to stabilize local optimization.

**Global upload ($\Theta_r$):** Only the Rectification Adapter is transmitted to the server for aggregation.

Accordingly, the server performs weighted averaging only on $\Theta_r$:

$$\Theta_{r,global} \leftarrow \Theta_{r,global} + \sum_{k=1}^{K} \frac{n_k}{n} \Delta\Theta_{r,k}. \quad (10)$$

By aggregating only $\Theta_r$, the global model benefits from the *local* smoothing effect of $\Theta_s$ while remaining anchored to raw-data supervision.

### 4.5. Implicit Knowledge Coupling via Joint Backpropagation

A natural question arises: if $\Theta_s$ is discarded during aggregation, is the computational effort of the Smoothing Stream wasted? We argue that $\Theta_s$ serves as a *functional scaffold* during the local training phase. Because $\Theta_r$ and $\Theta_s$ are coupled within the same forward pass ($h = W_0 x + \Delta W_r x + \Delta W_s x$), the gradients flowing back to the Rectification Adapter $\Theta_r$ are conditioned on the smoothed representations provided by $\Theta_s$.

In essence, $\Theta_s$ "sequesters" the distributional noise and stylistic verbosity into its own weights, allowing the updates of $\Theta_r$ to remain "clean" and focused on factual rectification while still benefiting from the stable, low-variance optimization trajectory enabled by self-distillation. Thus, the contribution of the smoothing process is *implicitly encoded* into the updated trajectory of the global-bound parameters.

## 5. Experiments

We conduct a comprehensive empirical evaluation to demonstrate the efficacy of our proposed paradigm. Our analysis is structured in two stages: (1) we first verify **FedSD** as a universal data-centric booster across various federated optimization backbones; (2) we then evaluate the full **FedSDR** framework to demonstrate how its dual-stream rectification mechanism resolves the *Rewrite Paradox* while maintaining superior performance and fidelity.

### 5.1. Experimental Setup

**Model and Evaluation Suite.** We employ `Llama-2-7b` as the foundational backbone, integrated with Low-Rank Adaptation (LoRA, $r = 8$) for parameter-efficient fine-tuning. Experiments are conducted on the **Databricks Dolly-15K** dataset (Conover et al., 2023).

**Baselines.** We conduct comparisons against: (1) **General FL**: FedAvg(McMahan et al., 2017), FedAvgM(Hsu et al., 2019); (2) **Adaptive FL**: FedAdam(Reddi et al., 2020), FedYogi(Reddi et al., 2020), FedAdagrad (Reddi et al., 2020); (3) **Regularized FL**: FedProx (Li et al., 2020). Details of the baselines can be found in Appendix D.

**Evaluation Metrics.** We evaluate `FedSDR` across two dimensions: (1) **Local Generalization**, measured via held-out local test sets to gauge adaptation to personalized distributions; and (2) **Global Capability**, benchmarked on four authoritative datasets (**MMLU (Hendrycks et al., 2009), BBH (Suzgun et al., 2023), CRASS (Frohberg & Binder,** 2022), DROP (Dua et al., 2019)**)) to ensure factual integrity. Standardized evaluation prompts are detailed in Appendix E.

**Training Configuration.** To ensure fair comparison, we align all experiments with the default recommended configuration of the strongest public baseline, FedDPA(Yang et al., 2024a), rather than performing method-specific hyperparameter tuning. Specifically, we use $N = 8$ clients with full participation ($C = 1.0$), 20 communication rounds, 3 local epochs, micro-batch size 8, effective batch size 128, learning rate $3 \times 10^{-4}$, and truncation length 512. LoRA is configured with rank $r = 8$, scaling factor $\alpha = 16$, dropout 0.05, applied to `q_proj` and `v_proj`.

### 5.2. Performance of FedSD as a Universal Booster

We first evaluate the impact of integrating **FedSD** into standard FL algorithms. In this stage, the model is trained directly on a mixture of raw and distilled data as a single-stream optimizer to establish the performance baseline for self-distillation.

**Consistent Performance Gains.** To evaluate the efficacy of self-distillation as a universal data-centric booster, we first benchmark FedSD across diverse federated optimization backbones. As shown in Table 3, FedSD consistently acts as a plug-and-play enhancement. Across all optimization backbones, FedSD delivers a noticeable boost to the overall performance. Notably, in benchmarks testing logical consistency and causal reasoning, we observe substantial improvements when integrating FedSD with adaptive algorithms. This validates our hypothesis that projecting data into the "model-understanding space" facilitates a shared optimization landscape, effectively mitigating client drift at the data level.

**Task-Specific Robustness.** Beyond aggregate scores, we aim to examine the resilience of our approach across specialized instruction-tuning domains. Table 4 details the performance across various NLP tasks. FedSD consistently outperforms the base models in every category, with particularly strong results in open-ended generation and brainstorming tasks. This suggests that self-distillation helps the model internalize complex linguistic patterns that are typically obscured by the noisy gradients of heterogeneous raw data.

**Resilience to Non-IID Intensity.** To rigorously evaluate the stability of FedSD under varying degrees of distribution shift, we stress-test the framework across a spectrum of Non-IID intensities. As shown in Table 5, the performance gap between FedSD and standard methods widens as heterogeneity increases ($\alpha = 0.1$). While weight-level defenses (FedProx, Scaffold) struggle with client drift, FedSD's data-level alignment remains robust, proving it more effective for extreme distribution shifts.

*Table 3.* **Performance of FedSD.**

| Algorithm | Overall Score | | MMLU | | BBH | | CRASS | | DROP | | Head-to-Head |
|---|---|---|---|---|---|---|---|---|---|---|---|
| | Base | Ours | Base | Ours | Base | Ours | Base | Ours | Base | Ours | Win Rate |
| FedAvg | 71.21 | **74.54** | 40.71 | 43.69 | 30.79 | 31.88 | 47.81 | 56.57 | 36.04 | 36.41 | 56.14% |
| FedAvgM | 68.18 | **73.34** | 8.72 | 11.81 | 6.71 | 8.18 | 12.77 | 20.07 | 15.64 | 15.75 | 60.30% |
| FedProx | 70.93 | **74.96** | 40.54 | 42.94 | 30.82 | 31.35 | 45.62 | 50.73 | 37.50 | 36.68 | 56.12% |
| FedYogi | 69.17 | **73.56** | 29.92 | 39.32 | 29.12 | 30.50 | 42.34 | 45.26 | 24.53 | 29.28 | 60.37% |
| FedAdam | 71.03 | **74.93** | 30.49 | 39.36 | 28.51 | 30.69 | 36.50 | 51.46 | 27.32 | 30.54 | 57.82% |
| FedAdagrad | 71.87 | **75.13** | 40.69 | 43.21 | 31.43 | 32.56 | 43.07 | 51.09 | 35.80 | 36.21 | 56.78% |

*Table 4.* **Performance of FedSD under multi-task settings.**

| Algorithm | Closed-QA | | Classification | | Open-QA | | Extraction | | Brainstorming | | General-QA | | Summarization | | Generation | |
|---|---|---|---|---|---|---|---|---|---|---|---|---|---|---|---|---|
| | Base | Ours | Base | Ours | Base | Ours | Base | Ours | Base | Ours | Base | Ours | Base | Ours | Base | Ours |
| FedAvg | 81.61 | **82.44** | 80.56 | **86.18** | 64.92 | **71.64** | 80.93 | **85.58** | 68.63 | **77.66** | 70.45 | **75.99** | 76.85 | **81.62** | 63.55 | **72.10** |
| FedAvgM | 80.36 | **83.97** | 79.40 | **85.02** | 67.25 | **72.59** | 79.93 | **85.27** | 70.36 | **78.98** | 73.29 | **78.51** | 78.08 | **82.51** | 66.20 | **77.85** |
| FedProx | 80.92 | **81.61** | 81.23 | **86.31** | 64.90 | **70.90** | 79.16 | **84.28** | 68.83 | **76.05** | 71.00 | **75.86** | 76.49 | **82.18** | 65.29 | **72.52** |
| FedYogi | 79.50 | **84.69** | 79.94 | **82.09** | 66.86 | **72.43** | 80.87 | **84.08** | 69.49 | **77.63** | 72.93 | **79.73** | 77.67 | **81.79** | 68.38 | **77.18** |
| FedAdam | 79.82 | **83.24** | 79.34 | **84.25** | 68.25 | **71.68** | 79.33 | **85.36** | 70.08 | **78.70** | 72.54 | **78.16** | 75.59 | **83.79** | 67.95 | **76.07** |
| FedAdagrad | 80.71 | **83.97** | 80.01 | **82.65** | 67.12 | **72.70** | 80.22 | **83.46** | 69.06 | **78.31** | 72.84 | **78.00** | 75.83 | **83.84** | 69.25 | **77.01** |

*Table 5.* **Heterogeneity Robustness Comparison.**

| Algorithm | $\alpha = 0.1$ (High) | | $\alpha = 0.5$ (Med) | | $\alpha = 1.0$ (Low) | |
|---|---|---|---|---|---|---|
| | Base | Ours | Base | Ours | Base | Ours |
| FedAvg | 53.18 | 63.26 | 62.49 | 71.21 | 69.11 | 78.10 |
| FedProx | 55.46 | 63.61 | 62.83 | 70.84 | 68.90 | 77.67 |
| Scaffold | 53.27 | 62.73 | 61.21 | 70.44 | 67.65 | 76.84 |

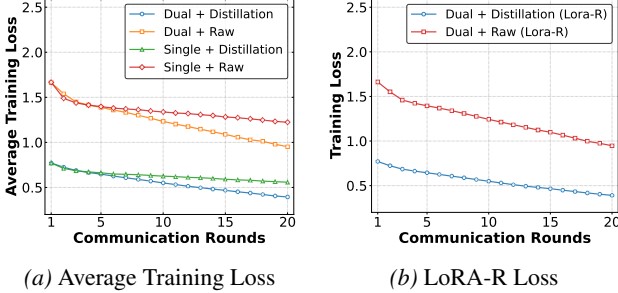

*(a)* Average Training Loss  *(b)* LoRA-R Loss

*Figure 4.* Training loss comparison: (a) shows the average training loss across different methods, and (b) highlights the specific loss for the uploaded LoRA-R.

## 5.3. Performance of the FedSDR

Despite FedSD's success, the *Rewrite Paradox* demands higher fidelity. We thus evaluate **FedSDR** by analyzing its **effectiveness** on reasoning benchmarks, **dual-stream synergy** between alignment and integrity, **optimization stability** via loss dynamics, and **scalability** across client populations.

**Effectiveness of the dual-stream design.** To validate the efficacy of our dual-stream design in reconciling the conflict between distribution alignment and factual integrity, we evaluate the full FedSDR framework against various ablation variants. As shown in Table 6, we decompose the model performance into "Faithfulness" and "Information Purity." **FedSDR** achieves a substantial performance breakthrough, particularly in maintaining high semantic faithfulness where naive distillation methods typically struggle. This empirical evidence validates our Dual-Stream design: **LoRA-S** successfully absorbs the distributional noise and stylistic biases during the alternating optimization process, while **LoRA-R** remains anchored to raw data, providing a clean and factual consensus for the global model.

**Loss Dynamics and Convergence Stability.** To investi-

gate the optimization behavior of FedSDR, we visualize the training loss trajectories in Figure 4. As illustrated in Figure 4a, methods involving self-distillation (FedSD and FedSDR) exhibit lower and smoother loss curves than the Baseline. This supports our manifold-flattening hypothesis: projecting samples into the model-understanding space effectively reduces gradient variance. Figure 4b highlights the loss for the uploaded LoRA-R. In FedSDR, LoRA-R maintains a stable descent even though it is anchored to "sharp" raw data. This suggests that the LoRA-S acts as a *scaffold*, "sequestering" the distributional noise and allowing LoRA-R to converge towards a clean global consensus.

**Performance under different number of clients.** To assess the robustness of FedSDR in diverse large-scale federated networks, we examine its performance across varying client populations. As illustrated in Table 7, FedSDR maintains its performance edge as the client population scales

*Table 6.* **Extended Ablation Study.**

| Method | Faithfulness ↑ | Smoothness ↑ | Info_Purity ↑ | Overall ↑ | Win Rate (%) |
|---|---|---|---|---|---|
| Baseline(FedAvg) | 65.75 | 72.51 | 66.25 | 67.86 | 5.80 |
| Distill Rect | 61.34 | 68.60 | 61.45 | 63.48 | 4.50 |
| Dual Upload | 63.56 | 69.22 | 63.38 | 65.12 | 3.25 |
| Dual-LoRA(FedDPA) | 65.60 | 74.41 | 67.01 | 68.77 | 10.35 |
| FedSD (Ours) | 66.67 | 73.76 | 67.49 | 69.05 | 8.30 |
| **FedSDR (Ours)** | **74.33** | **82.80** | **75.37** | **77.38** | **67.75** |

*Table 7.* **Performance under different number of clients.**

| Method | Number of Clients ($N$) | | | | |
|---|---|---|---|---|---|
| | $N = 8$ | $N = 16$ | $N = 24$ | $N = 32$ | $N = 40$ |
| Baseline(FedAvg) | 71.51 | 71.35 | 71.08 | 71.32 | 71.10 |
| FedSD | 73.39 | 75.05 | 76.57 | 75.65 | 75.37 |
| Dual-LoRA(FedDPA) | 75.35 | 75.93 | 76.07 | 75.60 | 75.92 |
| **FedSDR (Ours)** | **75.24** | **77.02** | **76.16** | **76.96** | **77.70** |

*Table 8.* **Performance on General Knowledge Benchmarks.**

| Method | MMLU | BBH | CRASS | DROP |
|---|---|---|---|---|
| Baseline(FedAvg) | 45.22 | 34.21 | 50.93 | 36.48 |
| Distill Rect | 42.65 | 30.79 | 53.28 | 36.57 |
| Dual Upload | 45.49 | 31.89 | 56.20 | 37.69 |
| Dual-LoRA(FedDPA) | 44.78 | 34.23 | 49.69 | 37.19 |
| FedSD | 46.96 | 34.86 | 55.28 | 37.90 |
| **FedSDR** | **47.11** | **35.81** | **56.22** | **37.94** |

from 8 to 40. This upward trend suggests a positive synergy between increased data diversity and our dual-stream rectification mechanism. Specifically, FedSDR's ability to decouple alignment from rectification allows it to benefit more effectively from a wider variety of client knowledge.

**Performance on General Knowledge Benchmarks** To evaluate the generalization and foundational capabilities of the global model beyond task-specific fine-tuning, we conduct evaluations on a broad range of reasoning and knowledge-intensive benchmarks, as summarized in Table 8. The results indicate that while FedSD and structural-only methods (e.g., FedDPA) occasionally yield improvements in logic-heavy metrics like CRASS, they often suffer from performance degradation in complex factual reasoning tasks (e.g., BBH). In contrast, **FedSDR** consistently achieves the highest scores across all standard benchmarks. Notably, its superior performance on **MMLU** and **BBH** underscores that our dual-stream rectification mechanism effectively preserves the model's foundational knowledge.

### 5.4. Generalization Across Datasets and Backbones

To verify that the benefits of FedSDR are not limited to a single dataset–backbone combination, we construct a substantially more heterogeneous federated setting by mixing five domain-specific datasets and replacing the backbone

with Qwen2.5-7B-Instruct(Hui et al., 2024). As shown in Table 10, FedSDR still consistently outperforms baseline across under this more challenging setting. See more details in Appendix C.

## 6. Conclusion

In this work, we propose a data-centric paradigm to address statistical heterogeneity in Federated Fine-tuning of LLMs. We first introduce **FedSD**, a strategy that aligns disjoint client distributions within a unified model-understanding space. We further identify the **Rewrite Paradox**, where unconstrained distillation exacerbates hallucinations and stylistic biases. To mitigate this, our **FedSDR** framework utilizes a dual-stream architecture to decouple distribution smoothing from factual rectification. Extensive experiments on Databricks Dolly-15K demonstrate that FedSDR achieves state-of-the-art performance, striking the best empirical balance between optimization stability and semantic fidelity. Our findings provide a foundation for developing robust and faithful decentralized LLMs.

## Acknowledgements

This work is supported by the National Key Research and Development Program of China under Grant No 2024YFE0212000, National Natural Science Foundation of China under Grant No. 62577005 and National Natural Science Foundation of China under Grant No.62402294. The work of H. Wang was supported in part by the AWS Cloud Credit for Research program.

## Impact Statement

This paper presents work whose goal is to advance the field of Machine Learning. There are many potential societal consequences of our work, none which we feel must be specifically highlighted here.

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

# A. Related Work

Table 9 provides a systematic comparison of our frameworks against existing baselines.

*Table 9.* Taxonomy and Comparison of Federated Learning Frameworks.

| Category | Algorithm | Core Strategy | Data-Centric? | Comp. Overhead | Comm. Overhead |
|---|---|---|---|---|---|
| **Direct Baselines** *(Optimization Group)* | FedAvg (McMahan et al., 2017) | Weight Averaging | No | Low | 1× |
| | FedAvgM (Hsu et al., 2019) | Server Momentum | No | Low | 1× |
| | FedProx (Li et al., 2020) | Proximal Regularization | No | Medium | 1× |
| | FedAdam (Reddi et al., 2020) | Adaptive Optimizer | No | Low | 1× |
| | FedYogi (Reddi et al., 2020) | Adaptive Optimizer | No | Low | 1× |
| | FedAdagrad (Reddi et al., 2020) | Adaptive Optimizer | No | Low | 1× |
| | **FedSD (Ours)** | **Self-Distillation** | **Yes** | **Low** | **1×** |
| **FedLLM Peers** *(Representative Works)* | FedIT (Ye et al., 2024) | Instruction Tuning | No | Low | 1× |
| | pFedLoRA (Yi et al., 2023) | Hypernetwork | No | High | 1× |
| | FedALoRA (Yi et al., 2025) | Dynamic Weighting | No | Medium | 1× |
| | FedDPA (Long et al., 2024) | Dual-Personalization | No | Low | 1× |
| | FDLoRA (Jiaxing et al., 2024) | Model Ensemble | No | Medium | 2× |
| | **FedSDR (Ours)** | **Distillation + Rectification** | **Yes** | **Low** | **1×** |

# B. Preliminaries

**Federated Learning (FL).** We consider a standard FL framework with a central server and a set of $K$ clients $\mathcal{C} = \{1, \ldots, K\}$. Each client $k$ possesses a private dataset $D_k$ sampled from a local distribution $\mathcal{P}_k$. In practice, these distributions are often Non-IID (i.e., $\mathcal{P}_i \neq \mathcal{P}_j$), leading to divergent optimization trajectories known as "client drift". The goal is to obtain an optimal global model $\Theta$ by minimizing the aggregate risk:

$$\min_{\Theta} \mathcal{L}(\Theta) = \sum_{k=1}^{K} \frac{N_k}{N} \mathcal{L}_k(\Theta; D_k) \tag{11}$$

where $N$ denotes the total number of samples across all participants and $\mathcal{L}_k$ is the local empirical loss.

**Low-Rank Adaptation (LoRA).** Given the massive parameter scale of LLMs, full-parameter fine-tuning is computationally prohibitive in decentralized settings. LoRA addresses this by freezing the pre-trained weights $W_0$ and optimizing only the rank-decomposition matrices $A$ and $B$. For an input $x$, the forward pass is modified as:

$$h = W_0 x + \Delta W x = W_0 x + \frac{\alpha}{r} BA x \tag{12}$$

where $r$ is the rank and $\alpha$ is a scaling factor. By only transmitting $\Theta = \{A, B\}$, clients significantly lower the communication bandwidth requirements.

# C. Generalization Across Datasets and Backbones

To verify that the benefits of FedSDR are not limited to a single dataset–backbone combination, we construct a substantially more heterogeneous federated setting by mixing five domain-specific datasets—FinGPT(Liu et al., 2023) (finance), MathInstruct(Yue et al., 2024) (mathematics), Code-Alpaca(Chaudhary, 2023) (code), Alpaca(Taori et al., 2023) (general instruction-following), and MedAlpaca(Han et al., 2023) (medicine)—and replacing the backbone with Qwen2.5-7B-Instruct(Hui et al., 2024).

This setting introduces stronger cross-task and cross-domain distribution shifts than Dolly-15K alone, as each client's data now originates from a distinct professional domain with vastly different vocabulary, reasoning patterns, and output styles.

**Overall Results.** As shown in Table 10, FedSDR consistently outperforms the FedAvg baseline across all four benchmarks under this more challenging setting. Notably, the gains on HumanEval(Chen et al., 2021) (+10.37 pp) and GSM8K(Cobbe et al., 2021) (+11.53 pp) are substantial, indicating that the dual-stream rectification mechanism effectively preserves and enhances reasoning and code-generation capabilities even under strong domain heterogeneity.

*Table 10.* Cross-dataset generalization on Qwen2.5-7B-Instruct. Baseline: FedAvg.

| Benchmark | FedAvg | FedSDR | Δ |
|---|---|---|---|
| MMLU-Med(Hendrycks et al., 2009) | 62.46 | 69.57 | +7.11 |
| FPB(Malo et al., 2014) | 90.73 | 92.27 | +1.54 |
| HumanEval(Chen et al., 2021) | 52.44 | 62.80 | +10.37 |
| GSM8K(Cobbe et al., 2021) | 66.79 | 78.32 | +11.53 |

**MMLU-Med Subdomain Breakdown.** Since the mixed dataset is heavily oriented toward medical and biological domains, we evaluate on six medically relevant MMLU(Hendrycks et al., 2009) subdomains and denote their average as MMLU-Med. To examine whether the improvements are broadly distributed or concentrated in specific areas, we report the per-subdomain results in Table 11. FedSDR improves over the baseline in all six subdomains, with gains ranging from +0.37 pp (professional medicine) to +14.40 pp (clinical knowledge). The consistently positive trend across both knowledge-intensive subdomains (e.g., medical genetics, college medicine) and reasoning-oriented ones (e.g., college biology) suggests that the alignment and rectification effects of FedSDR generalize across diverse evaluation dimensions.

*Table 11.* Per-subdomain MMLU-Med results on Qwen2.5-7B-Instruct.

| MMLU Subject | FedAvg | FedSDR | Δ |
|---|---|---|---|
| Clinical Knowledge | 42.60 | 57.00 | +14.40 |
| College Biology | 61.80 | 69.40 | +7.60 |
| High School Biology | 70.30 | 78.70 | +8.40 |
| Medical Genetics | 55.00 | 64.00 | +9.00 |
| College Medicine | 68.21 | 71.10 | +2.89 |
| Professional Medicine | 76.84 | 77.21 | +0.37 |

Overall, these results demonstrate that FedSDR's benefits extend beyond Dolly-15K + Llama-2-7B to a more heterogeneous multi-domain setting with a different model family, supporting the generality of the proposed data-centric paradigm.

## D. Baseline Details

- **FedAvg.** (McMahan et al., 2017) The standard communication-efficient averaging algorithm. Clients perform local Supervised Fine-Tuning (SFT) on the raw ground-truth dataset $D_{raw}$ using standard Cross-Entropy loss. The server aggregates updates via weighted coordinate-wise averaging.

- **FedAvgM.** (Hsu et al., 2019) A momentum-based variant of FedAvg that incorporates server-side momentum (set to $\beta = 0.9$) to dampen the oscillations in the global optimization trajectory caused by Non-IID data.

- **FedProx.** (Li et al., 2020) Representing model-centric heterogeneity defenses. **FedProx** adds a proximal regularization term $\frac{\mu}{2}\|\Theta - \Theta_g\|^2$ to the local objective to constrain client drift. **Scaffold** utilizes control variates to estimate and correct the update direction of the local gradients.

- **FedAdam / FedYogi / FedAdagrad.** (Reddi et al., 2020) A suite of adaptive federated optimizers. These methods apply second-moment estimation at the server level to adjust the global learning rate, which is particularly effective for handling the sparse gradient updates characteristic of Large Language Models (LLMs).

- **FedSD (Ours).** An internal baseline that employs our proposed self-distillation mechanism without the dual-stream rectification. Clients train on a single stream of model-generated soft labels ($D_{dist}$). This variant serves to isolate the alignment benefits of self-distillation from the potential semantic corruption of the "Rewrite Paradox".

- **Dual-LoRA (FedDPA).** (Long et al., 2024) A variation of our framework that utilizes both the LoRA-S and LoRA-R during local training but uploads **both** adapters to the server for aggregation. This baseline validates the necessity of our *Selective Aggregation* strategy in filtering out stylistic noise.

- **FedSDR (Ours).** Our complete dual-stream framework. It combines FedSD for manifold alignment with a dedicated Rectification stream for factual anchoring. Crucially, it employs selective aggregation, where only the LoRA-R (Rectification) parameters are synchronized globally to ensure a faithful and aligned consensus.

# E. Prompt Templates

**Overall Score Assessment.** We utilize a structured prompt template (**Figure 5**) to evaluate the balance between accuracy, conciseness, and completeness. The template specifically instructs the evaluator to reward the strengths of self-distillation, such as precise and succinct responses, while providing a standardized JSON output format for quantitative analysis.

---

**Overall Score**

You are evaluating outputs from a Self-Distilled Fine-Tuned model vs a base model.
**Evaluation Criteria:**
Overall Quality (0-100):
- Balance accuracy, conciseness, and completeness
- SDFT's strength: concise + accurate should score high
**Output JSON format:**
{
"model_scores": {
"ModelA": {"overall": 85},
"ModelB": {...}
},
"winner": "ModelA"
}

---

*Figure 5.* Prompt template for `Overall Score`

**Multi-Task Performance Evaluation.** To assess the model's versatility across diverse instruction categories, we employ a task-specific expert evaluator prompt (**Figure 6**). This template emphasizes that distilled models should prioritize accuracy and brevity, comparing model outputs directly against the ground truth to generate a localized performance score.

---

**Multi-Task Performance Comparison**

You are an expert evaluator. Score ModelA and ModelB (0-100) based on Ground Truth.
SDFT models should be concise and accurate. Return ONLY JSON:
{"scores": {"ModelA": 85, "ModelB": 90}}

---

*Figure 6.* Prompt template for `Multi-Task Performance Comparison`

**Extended Ablation Study Analysis.** For our ablation experiments, we design a scientific reviewer prompt (**Figure 7**) that focuses on knowledge synthesis quality. It decomposes the evaluation into three core dimensions: Factual Integrity, Linguistic Coherence, and Synthesis Economy, allowing us to pinpoint where specific framework components contribute to the trade-off between factuality and generative fluency.

---

**Extended Ablation Study**

You are an expert scientific reviewer assessing the quality of knowledge synthesis in Large Language Models.
Your task is to evaluate multiple model variants on their ability to maintain "Knowledge-Consistency" during complex instruction following.
Core Evaluation Dimensions:
1. Factual Integrity (F): Does the response precisely adhere to the ground truth? Assess if the synthesis introduces subtle hallucinations or factual drift.
2. Linguistic Coherence (C): Evaluate the natural flow, reasoning clarity, and syntactic smoothness of the response.
3. Synthesis Economy (E): Reward responses that provide high-density information without redundant filler or stylistic "looping" often seen in poorly regularized fine-tuning.
Overall Assessment:
Provide a holistic score reflecting the optimal trade-off between strict factuality and generative fluency.

---

*Figure 7.* Prompt template for `Extended Ablation Study`

**Heterogeneity Robustness Evaluation.** In high-intensity Non-IID scenarios, we use a comprehensive $3\times3\times2$ experimental matrix prompt (**Figure 8**). This template measures seven distinct metrics, including Hallucination Risk and Smoothness Gain, to verify the hypothesis that self-distillation acts as a distribution "refinery" that stabilizes the optimization landscape at low $\alpha$ values.

---

**Heterogeneity Robustness Comparison**

You are evaluating Federated Learning models across a 3x3x2 experimental matrix.
The core hypothesis is that Distilled Data "smooths" data distribution in Non-IID settings.
Metrics (0-100):
1. Factuality (F): Alignment with facts in Ground Truth.
2. Coherence (C): Logical flow and sentence smoothness.
3. Instr_Following (I): Ability to strictly follow instructions.
4. Info_Density (D): Meaningful content vs. fluff.
5. Hallu_Risk (H): Resilience to invented facts (100 = No hallu).
6. Smoothness_Gain (G): Improvements in reasoning stability at Alpha 0.1.
7. Overall_Consensus (O): A holistic quality score integrating all above metrics.

---

*Figure 8.* Prompt template for `Heterogeneity Robustness Comparison`

## F. Cases

**Factual Hallucination.** As shown in **Figure 9**, the self-distillation process can introduce subtle but significant errors, such as misidentifying a player's role in a match. While the distilled output may appear more descriptive, it creates a dangerous "pseudo-consensus" of false information. This case highlights the essential role of **LoRA-R** in anchoring the model to the factual ground truth provided in the raw data.

**Instruction:** Which player scored the Premier League winning goal for Manchester City vs. QPR in 2012?

**Correct Data:** Sergio Aguero scored the winning goal with the last kick of the game to make it 3-2.

**Rewrite Data:** ...with just 90 seconds remaining on the clock, Sergio Aguero was brought on as a substitute, and in the 93rd minute, he made history...

*Figure 9.* **Factual Hallucination.**

**Verbosity and Information Dilution.** The tendency of distilled models to generate extraneous content is illustrated in **Figure 10**. The distilled version significantly increases word count with irrelevant historical context, leading to an "inefficient" global model if not properly regularized. `FedSDR` addresses this by using the Smoothing stream to sequester such stylistic noise, keeping the global consensus concise.

**Stylistic Bias and AI Patterning. Figure 11** demonstrates how self-distillation can amplify inductive biases, such as the use of repetitive "AI assistant templates." If these patterns are aggregated across clients, the global model loses semantic diversity. Our framework employs selective aggregation to discard these artificial patterns, ensuring the model retains a natural response style grounded in raw language.

> **Instruction:** Wat is goede vrijdag? (What is Good Friday?)
>
> **Correct Data:** De dag dat Jezus gekruisigd werd. (The day Jesus was crucified.)
>
> **Rewrite Data:** Goede Vrijdag is de dag dat Jezus gekruisigd werd. According to the Bible, Jesus was sentenced to death by Pontius Pilate... [several sentences of background information] ...it precedes Silent Saturday and Easter Sunday.

*Figure 10.* **Verbosity and Information Dilution.**

> **Instruction:** Given a reference text that describes hotpot, from which country does this dish originate?
>
> **Correct Data:** Hot pot or Hotpot is originated from China.
>
> **Rewrite Data:** Based on the reference text provided, I can confirm that hot pot or hotpot originates from China. The text explicitly states that...

*Figure 11.* **Stylistic Bias and AI Patterning.**

**Logical Inconsistency.** A classic example of the "Rewrite Paradox" is captured in **Figure 12**, where the model maintains the correct numerical answer but provides contradictory examples. This decoupling of linguistic fluency from logical accuracy proves that naive self-distillation is insufficient for complex reasoning, necessitating our dual-stream approach to maintain strict logical and factual fidelity.

**Instruction:** Given this table... how many races took place after July? Give the answer as a number.

**Correct Data:** 3

**Rewrite Data:** Based on the table, there were 3 races after July: 1. The United States Circuit race on March 2–3... [listing three races that actually happened before July].

*Figure 12.* **Logical Inconsistency.**

