# OpenReview forum: "FedSDR: Federated Self-Distillation with Rectification"
_ICML.cc/2026/Conference — ICML 2026 regular_

### Official Review · Reviewer_rKhU · 2026-03-12

**Soundness:** 3
**Presentation:** 3
**Significance:** 3
**Originality:** 3
**Overall Recommendation:** 4
**Confidence:** 3

**Summary:**

This paper proposes FedSDR, a data-centric method for federated fine-tuning of large language models. It mitigates statistical heterogeneity by projecting disparate client data into a unified space via self-distillation, and employs a dual-LoRA architecture to decouple distribution smoothing from factual rectification, aggregating only the rectification stream parameters. Experiments show the method effectively addresses the hallucination and redundancy issues caused by self-distillation, achieving state-of-the-art performance on multiple benchmarks.

**Compliance With Llm Reviewing Policy:**

Affirmed.

**Key Questions For Authors:**

Please refer to the Weaknesses part.

**Limitations:**

Yes

**Strengths And Weaknesses:**

Strengths:

1. Comprehensive experiments covering multiple FL algorithms, task types, and client scales.

2. Strong baselines (FedAvg, FedProx, FedAdam, etc.) provide compelling evidence.

3. Multi-dimensional evaluation (global capability, local generalization, task types) demonstrates the method's advantages.

Weaknesses:

1. The hallucination and redundancy issues introduced by self-distillation are only briefly illustrated in the Appendix, lacking quantitative metrics (e.g., hallucination rate, redundancy ratio) to substantiate the phenomenon.

2. It is unclear why only LoRA-R is aggregated. The paper claims LoRA-S "absorbs noise," but does not explain whether this noise would affect the global model. If LoRA-S truly absorbs noise, should it also be aggregated to improve generalization?

3. Several performance improvements are only 1-2 percentage points, with no statistical significance reported. Gains on benchmarks like MMLU and BBH are marginal, raising concerns about overfitting.

4. The definition of "data heterogeneity" remains at the label distribution level, without considering more complex scenarios such as semantic shift or task drift.

---

> ### Author Rebuttal · Authors · 2026-03-31
>
> Thank you for the constructive feedback. Below we respond point by point.
>
> > **Response to Weaknesses 1**
>
> Regarding the hallucination and redundancy issues, the main paper already provides quantitative evidence in Figure 3. Among distilled responses, only 48.0% strictly entail the original answers, 47.0% are neutral, and 5.0% are contradictions. Meanwhile, average response length increases to 1.24× that of raw answers, and filler-word frequency rises from 10.6/1k to 20.3/1k—indicating both factual drift and redundant elaboration. About 52% of distilled samples do not strictly entail the original answer, and rewriting makes responses longer and less information-dense.
>
> These statistics directly motivate LoRA-R. If rewritten supervision were uploaded in a single-stream framework, hallucination and stylistic bias could be amplified through federated aggregation. FedSDR prevents this: LoRA-S absorbs the smoothing benefit and rewrite noise locally, while LoRA-R remains anchored to raw data as the only uploaded branch—a structural rectification mechanism for the Rewrite Paradox verified in Figure 3.
>
> > **Response to Weaknesses 2**
>
> LoRA-S absorbs data heterogeneity but also inherits the Rewrite Paradox from Section 3.2, inevitably acquiring rewriting artifacts such as factual hallucinations, stylistic verbosity, and templated phrasing. We therefore treat LoRA-S as a local scaffold rather than a branch for global aggregation, as uploading it would propagate local rewrite biases into the global model.
>
> We test this through the Dual Upload baseline in Tables 4 and 6. In Table 4, Faithfulness drops to 63.56 under Dual Upload versus 74.33 under FedSDR. In Table 6, Dual Upload reaches only 45.49 on MMLU and 31.89 on BBH, while FedSDR improves to 47.11 and 35.81. Excluding LoRA-S thus protects the global model rather than sacrificing generalization.
>
> > **Response to Weaknesses 3**
>
> These results should be interpreted in the context of federated LLM fine-tuning rather than only through broad benchmarks such as MMLU or BBH.
>
> The most direct evaluation is the test set of federated task itself. Gains are clearly not limited to 1–2 points: in Table 2, FedSD improves over the baseline in all 48 results across 6 FL algorithms × 8 task categories, with an average gain of 5.72 points.
>
> MMLU and BBH are mainly tests of out-of-distribution capability retention. Because each client has limited local data, improvements on broad external benchmarks are naturally smaller. Even so, the trend remains consistently positive: in Table 1, the overall score improves for all 6 FL backbones, with an average gain of 4.01 points (max 5.16, min 3.26), and per-benchmark gains of +4.88 on MMLU, +1.30 on BBH, +7.85 on CRASS, and +1.34 on DROP.
>
> The results do not support an overfitting explanation: if the method overfit the training distribution, we would expect degraded performance on external benchmarks. Instead, consistent positive gains on both primary and general benchmarks suggest better heterogeneity mitigation rather than simple memorization.
>
> > **Response to Weaknesses 4**
>
> Interpreting "data heterogeneity" as classical label skew alone is insufficient for federated LLM fine-tuning. Our setting focuses on task-dependent and semantic heterogeneity from the outset.
>
> Databricks Dolly-15K spans multiple task types that vary greatly from each other, and partitioning clients by task category introduces not only sample proportion differences, but also task drift and semantic shift. For example, one client may focus on creative writing with open-ended generation, while another handles closed-QA with short, fact-oriented outputs, and clients can further differ in vocabulary, syntax, and generation style.
>
> Our Non-IID setup thus simulates task-dependent and semantic heterogeneity—which is why we analyze both textual and optimization statistics before and after self-distillation, targeting update conflicts across different tasks and semantic distributions.
>
> To modeling more complex scenarios, we further added a cross-dataset experiment mixing **FinGPT, MathInstruct, Code-Alpaca, Alpaca, and MedAlpaca** with backbone Qwen2.5-7B-Instruct, spanning finance, mathematics, code, general instruction-following, and medical domains. FedSDR still achieves consistent gains under this substantially stronger cross-task and cross-domain shift:
>
> | Benchmark | Baseline | Ours | Delta |
> |---|---:|---:|---:|
> | MMLU-Med | 0.6246 | 0.6957 | +0.0711 |
> | FPB | 0.9073 | 0.9227 | +0.0154 |
> | HumanEval | 0.5244 | 0.6280 | +0.1037 |
> | GSM8K | 0.6679 | 0.7832 | +0.1153 |
>
> These results suggest that the heterogeneity addressed by FedSDR extends beyond task labels within Dolly-15K to stronger cross-task, cross-domain, and cross-semantic-distribution shifts.

---

> > ### Author Rebuttal · Reviewer_rKhU · 2026-04-01
> >
> > The author's response partially answered my questions in both the theoretical and experimental parts, which is why I improved my score.

---

> > > ### Author Response · Authors · 2026-04-01
> > >
> > > Dear Reviewer,
> > >
> > > Thank you for reading our paper so carefully and giving clear, practical feedback! Your comments showed us what needed fixing and pushed us to add the right experiments. We're glad our answers helped and appreciate you raising the score. The paper is better because of your review.
> > >
> > > Thanks again,
> > >
> > > The authors

---

### Official Review · Reviewer_1usw · 2026-03-13

**Soundness:** 3
**Presentation:** 2
**Significance:** 3
**Originality:** 3
**Overall Recommendation:** 4
**Confidence:** 3

**Summary:**

This paper studied the problem of LLM fine-tuning under federated data heterogeneity. It highlighted that Federated Self-Distillation (FSD) enabled aligning the heterogeneous data distributions. FedSDR was further introduced to decouple distribution smoothing from factual rectification. Extensive experiments demonstrated the effectiveness of the proposed FedSDR framework in balancing optimization stability and semantic fidelity.

**Compliance With Llm Reviewing Policy:**

Affirmed.

**Key Questions For Authors:**

(1) How are the varying degrees of distribution shift in Table 3 obtained?

(2) There are some minor typos: e.g., "Performance under different nuber of clients" in Table 5.

**Limitations:**

The applications of the proposed framework in other data modalities can be discussed.

**Strengths And Weaknesses:**

Strengths:

(S1) A federated self-distillation (FSD) strategy is proposed to align disjoint client distributions within a unified space for handling the data heterogeneity in FL.

(S2) The FedSDR framework is introduced based on the dual-stream LoRA architecture to mitigate the hallucinations.

(S3) Extensive experiments demonstrated the effectiveness of the proposed FedSDR framework.

Weaknesses:

(W1) It is unclear why the rewriting strategy in Module 1 can align disjoint client distributions. More statistical properties of rewritten client data can be analyzed to verify the distribution alignment among clients.

(W2) The major concern is the reproducibility of the experiments. More details about the experimental setup can be provided. It is not explained why the data heterogeneity is produced across clients when using the Databricks Dolly-15K datasets.

(W3) The "optimal balance between optimization stability and semantic fidelity" of FedSDR in the conclusion section can be explained and verified. How is this balance measured? Why does it argue that FedSDR achieves the optimal balance compared to baselines?

---

> ### Author Rebuttal · Authors · 2026-03-31
>
> Thank you for the constructive suggestions! We respond to the questions (Q) and weaknesses (W) below.
>
> > **Response to W1**
>
> Actually, our t-SNE visualization in Figure 1 shows FedSD dissolves distinct client clusters into a shared, synchronized distribution to facilitate stable global aggregation. FedSD achieves this by mapping high-variance and stylistically diverse human supervision into the response distribution of the same teacher model, reducing superficial lexical/syntactic discrepancies and making client optimization objectives more compatible.
>
> To verify this, we provide both text-distribution and optimization-level evidence. At the text-distribution level, we measure inter-client divergence using **JS divergence** and **TF-IDF cosine similarity** before and after the self-distillation process:
>
> | Statistic | Change |
> |---|---:|
> | Average JS Divergence ↓ | −11.4% |
> | Average TF-IDF Cosine Similarity ↑ | +11.0% |
>
> Rewritten client texts become closer to one another, suggesting that rewriting compresses dispersed client expressions into a more unified representational space via the shared generative preference of the same teacher model.
>
> At the optimization level, we further added two analyses on five highly heterogeneous domain datasets (**FinGPT, MathInstruct, Code-Alpaca, Alpaca, and MedAlpaca**): **cross-task gradient cosine similarity** and **cross-task loss transfer**. We summarize for each source task the average scores over the other four target tasks. Here, **Grad. Sim.** denotes the cross-task gradient cosine similarity, and **Loss Trans.** denotes the average change in loss for other tasks before and after training on a certain source task:
>
> | Source task | Grad. Sim. | Loss Trans. |
> |---|---:|---:|
> | FinGPT | +41.5 pp | -2.5 pp |
> | MathInstruct | +13.8 pp | -4.0 pp |
> | Code-Alpaca | +13.6 pp | -4.9 pp |
> | Alpaca | +10.8 pp | -5.1 pp |
> | MedAlpaca | +8.1 pp | -5.4 pp |
>
> All five source tasks show positive gains in gradient similarity and negative loss changes, meaning update directions across tasks become more aligned and training on one task more consistently benefits others.
>
> > **Response to W2/Q1**
>
> Dolly-15K contains ~15k instruction-following samples spanning 8 task types that vary greatly from each other, suitable for simulating federated settings where clients prefer different tasks. Similar task-based non-IID constructions have been adopted in prior federated LLM work [1–3]. Heterogeneity comes from assigning different task categories or mixtures to different clients, inducing both feature shift and response-distribution shift.
>
> As to the modeling of the varying degrees of distribution shift in Table 3, we also adopt a generalized strategy [1-3] based on the **Dirichlet distribution**. We treat the 8 task categories in Dolly-15K as a set of labels. For each client $k$, we draw a probability vector $p_k \sim Dir(\alpha \cdot \mathbf{1})$, where $\mathbf{1}$ is the all-ones vector and $\alpha > 0$ is the concentration parameter. The vector $p_k = (p_{k,1}, \dots, p_{k,8})$ determines the proportion of samples from each task category assigned to client $k$. By adjusting $\alpha$, we can precisely control the sparsity of the resulting distribution:
> - `α = 0.1` (High): each client typically contains data from only 1–2 task types;
> - `α = 0.5` (Medium): moderately heterogeneous;
> - `α = 1.0` (Low): weakly heterogeneous, close to IID.
>
> For reproducibility, we will clarify the main setup in the appendix: 8 clients, 20 communication rounds, 3 local epochs, effective batch size 128, and learning rate 0.0003. To ensure fairness, we aligned experiments with the default recommended settings of strong baselines (FedDPA) rather than performing method-specific tuning (LoRA rank = 8, alpha = 16).
>
> > **Response to W3**
>
> This balance is reflected in: (1) **optimization stability**—strong robustness under highly heterogeneous settings (e.g., α = 0.1 in Table 3) and smoothed training loss in Figure 4; and (2) **semantic fidelity**—mitigation of the Rewrite Paradox after introducing LoRA-R, as shown by our fidelity-oriented analyses (Table 4) and stronger performance on fact-sensitive benchmarks such as MMLU and BBH (Table 6).
>
> FedSDR achieves this because its dual-stream design explicitly separates the two goals: LoRA-S absorbs noise and improves optimization stability, while LoRA-R remains anchored to raw data and preserves factual correctness. As shown in Tables 4 and 6, FedSDR is the only framework that performs best on both dimensions simultaneously.
>
> > **Response to Q2**
>
> Thank you for the careful reading. We have corrected the typo in Table 5 and will proofread the full manuscript to eliminate similar issues.
>
> ---
>
> [1] *Towards Building the Federated GPT: Federated Instruction Tuning*
> [2] *FedPT: Federated Proxy-Tuning of Large Language Models on Resource-Constrained Edge Devices*
> [3] *Federated Fine-tuning of Large Language Models under Heterogeneous Tasks and Client Resources*

---

> > ### Author Rebuttal · Reviewer_1usw · 2026-04-03
> >
> > Thanks for the rebuttal. Most of my questions have been well addressed. There are some concerns left after reading the response.
> >
> > (1) Regarding the reproducibility in "Response to W2/Q1", what are "the default recommended settings of strong baselines (FedDPA)"? Why does it enforce a fair comparison compared to method-specific tuning?
> >
> > (2) Regarding the "optimal balance between optimization stability and semantic fidelity" in Response to W3, it is argued that FedSDR is the only framework that performs best on both dimensions simultaneously. This makes the statement "optimal balance" between dimensions hard to follow. It seems FedSDR aims to improve both optimization stability and semantic fidelity simultaneously, rather than achieving an optimal balance/trade-off between them.

---

> > > ### Author Response · Authors · 2026-04-04
> > >
> > > Dear Reviewer,
> > >
> > > Thank you for your prompt response and for engaging so deeply with our work! We are glad that our previous response addressed most of your questions, and we appreciate the opportunity to clarify your remaining concerns.
> > >
> > > **(1) Regarding reproducibility and the rationale for a "fair comparison":**
> > > All of our relevant training, models, and data parameter settings are completely in line with the official open-source code of FedDPA [1]. As clarified in our "Response to W2/Q1" regarding the general optimization hyperparameters, we specifically utilized 8 clients, 20 communication rounds, 3 local epochs, a micro-batch size of 8, a client selection fraction of 1.0, an effective batch size of 128, a cutoff length of 512, a learning rate of 0.0003, and even the exact same prompt template. Furthermore, we also ensured completely identical LoRA parameters to avoid any method-specific tuning based on the LoRA module itself. Specifically, we set the LoRA rank = 8, alpha = 16, and lora_dropout = 0.05, applying them uniformly to the $W_q$ and $W_v$ projection matrices.
> > >
> > > The rationale for strictly adopting these settings without performing method-specific tuning for FedSDR is to eliminate hyperparameter optimization as a confounding variable. By intentionally adopting the exact settings optimized for our strongest baseline, we effectively evaluate FedSDR at a "home-field disadvantage." Because FedSDR consistently outperforms FedDPA even under FedDPA’s own **default recommended** configurations, it provides rigorous evidence that the performance gains stem fundamentally from our dual-stream, data-centric design. We will explicitly detail these baseline settings and this rationale in the reproducible setup section of the appendix.
> > >
> > > [1] https://github.com/Lydia-yang/FedDPA/tree/main
> > >
> > > **(2) Regarding the phrasing of "optimal balance":**
> > > We sincerely thank you for this sharp observation, which beautifully highlights and elevates the true impact of our framework. We deeply appreciate your recognition of our method's full potential that FedSDR goes beyond merely managing a traditional trade-off but achieves a simultaneous improvement on both fronts.
> > >
> > > Our original phrasing of "optimal balance" was intended to reference the inherent trade-off we observed in standard self-distillation (FedSD alone), where gaining optimization stability traditionally comes at the cost of semantic fidelity due to the "Rewrite Paradox." The explicit purpose of FedSDR's dual-stream architecture is to **break** this trade-off by decoupling the conflicting objectives.
> > >
> > > Thank you for pointing out this semantic imprecision; "simultaneously enhancing" is indeed a much more accurate and powerful description of our framework's achievement than "balancing." We will gladly revise relevant discussions in the manuscript to replace "striking an optimal balance" with "simultaneously enhancing both optimization stability and semantic fidelity by breaking the traditional distillation trade-off."
> > >
> > > Thank you again for your continued support and the highly constructive feedback you have provided, which has undoubtedly helped us strengthen the final version of our paper.
> > >
> > > Best regards,
> > >
> > > The Authors

---

### Official Review · Reviewer_ofrB · 2026-03-13

**Soundness:** 3
**Presentation:** 3
**Significance:** 2
**Originality:** 3
**Overall Recommendation:** 3
**Confidence:** 3

**Summary:**

This paper investigates federated fine-tuning of large language models under statistical heterogeneity and advocates a data-centric solution based on self-distillation. The authors first propose FedSD, which reformulates local supervision using the model’s own outputs, arguing that this helps map heterogeneous client data into a shared “model-understanding space.” They then identify what they call the “Rewrite Paradox”: naive self-distillation may lead to hallucinations, excessive verbosity, and stylistic bias. To address this issue, they introduce FedSDR, a dual-stream LoRA framework consisting of a smoothing branch trained on distilled data and a rectification branch trained on raw data, with only the rectification branch being aggregated across clients. Experiments on Dolly-15K, using several federated learning backbones and benchmark evaluations, show that the proposed method outperforms standard FL baselines as well as several ablation variants.

**Compliance With Llm Reviewing Policy:**

Affirmed.

**Key Questions For Authors:**

1. Theoretical justification. Can the authors provide any theoretical analysis of FedSDR, such as convergence guarantees, generalization bounds, or a formal explanation of heterogeneity reduction?
2. Stability under partial local updates. Relatedly, how does FedSDR avoid instability or drift caused by partial local updates across heterogeneous clients? Have the authors observed any failure cases or non-convergent behaviors during training？
3. Generality across models and datasets. Have the authors tested FedSDR on additional datasets or backbones beyond Dolly-15K and Llama-2-7b?
4. Missing experimental details. Several important experimental settings are unclear. Could the authors specify the number of training epochs, the communication period, and any other key optimization settings needed for reproducibility? Was hyperparameter grid search performed to obtain the best test-set performance for each baseline algorithm?
5. Human validation / robustness checks. Were any human evaluations conducted to validate the custom metrics in Table 4? If not, can the authors provide evaluator robustness checks, such as sensitivity to prompt wording, agreement across different LLM judges, or correlation with human preferences?

**Strengths And Weaknesses:**

Strengths：
1. The proposed idea is a sensible way to reduce client heterogeneity before or during federated optimization. This framing is interesting and, at least conceptually, offers a more data-centric perspective than the usual parameter-regularization approach.
2. The empirical results also show a fairly consistent pattern: self-distillation improves over the corresponding base FL optimizer, and the gains reported in Table 1 are non-trivial.

Weaknesses：
1. The paper does not provide any theoretical grounding for its performance claims. In particular, there is no convergence result, no generalization analysis, and no formal treatment of how the method reduces client heterogeneity. Given that the method updates only a subset of local parameters, some discussion of when and why the global model  still converge seems essential.
2. All of the training experiments are run on a single dataset-model combination: Databricks Dolly-15K with Llama-2-7B. That is too narrow an empirical basis to show the superiority of the proposed algorithms. It is highly necessary to conduct experiments on more datasets and more LLM-variants(such as Qwen3-4B/8B).
3. The experimental setup lacks important details, such as the number of training epochs, the communication period, and whether hyperparameter grid search was performed to obtain the best test-set performance for each algorithm.
4. The main evidence supporting FedSDR over FedSD and other ablations comes from Table 4, but its metrics—“Faithfulness,” “Smoothness,” “Info_Purity,” “Overall,” and “Win Rate”—are custom judged scores. The appendix prompt templates raise further concern by seemingly encoding the authors’ preferred behavior, such as favoring concise distilled outputs. Without human validation or evaluator robustness checks, these results are not strong enough to support the paper’s central claims.

---

> ### Author Rebuttal · Authors · 2026-03-31
>
> Thank you for the careful reading! Below we respond your questions and weaknesses point by point.
>
> > **Response to Weakness1 / Question 1&2**
>
> Since our work is to supplement the current all-classical model-centric fl algorithms from the perspective of the data distribution level, without modifying any model architecture, and all the theoretical convergence proofs of the related frameworks are data-agnostic and can serve as our theoretical foundation.  Regarding the issue where the model only updates a subset of local parameters (LoRA), we mainly adopted the setting of the previous mainstream federated learning work for LLMs[1,2] (mainly considering the computational and communication costs brought by LLMs); The related empirical convergence analysis has also been thoroughly supported by previous studies[1,2]. Therefore, we do not elaborate on these contents in the paper. In our paper, Figure 4 also shows that both the **global average training loss** and the uploaded **LoRA-R** loss decrease smoothly over communication rounds, with no unstable or non-convergent behavior observed in practice.
>
> To more directly prove self-distillation aligns heterogeneous data, we further added two analyses on five highly heterogeneous domain datasets (**FinGPT, MathInstruct, Code-Alpaca, Alpaca, and MedAlpaca**): **cross-task gradient cosine similarity** and **cross-task loss transfer**.
> We summarize for each source task the average scores over the other four target tasks. Here, **Grad. Sim.** denotes the cross-task gradient cosine similarity, and **Loss Trans.** denotes the average change in loss for other tasks before and after the model was trained on a certain source task. Below are the rates of change of these two metrics after the self-distillation process:
>
> | Source task | Grad. Sim. | Loss Trans. |
> |---|---:|---:|
> | FinGPT | +41.5 pp | -2.5 pp |
> | MathInstruct | +13.8 pp | -4.0 pp |
> | Code-Alpaca | +13.6 pp | -4.9 pp |
> | Alpaca | +10.8 pp | -5.1 pp |
> | MedAlpaca | +8.1 pp | -5.4 pp |
>
> All five source tasks show positive gains in gradient similarity and negative loss changes, meaning that after self-distillation, update directions across tasks become more aligned and training on one task more consistently benefits others. Overall, these analyses provide mechanism-level support for our interpretation: self-distillation **reshapes heterogeneous supervision into a more aligned and cooperative optimization space**.
>
> > **Response to Weakness 2 / Question 3**
>
> To address the concern of experimental scope, we added experiments across both data distributions and model families. We use the new mixed domain dataset mentioned above, replaced the backbone with Qwen2.5-7B-Instruct, and evaluate on corresponding domain benchmarks. FedSDR still achieves consistent gains in this more heterogeneous setting:
>
> | Benchmark | Baseline | Ours | Delta |
> |---|---:|---:|---:|
> | MMLU-Med | 0.6246 | 0.6957 | +0.0711 |
> | FPB | 0.9073 | 0.9227 | +0.0154 |
> | HumanEval | 0.5244 | 0.6280 | +0.1037 |
> | GSM8K | 0.6679 | 0.7832 | +0.1153 |
>
> > **Response to Weakness 3 / Question 4**
>
> To ensure a fair comparison, we aligned our settings with the default recommended configuration of the strongest public baseline, FedDPA[2], rather than performing method-specific hyperparameter tuning. Our main training setup is: 8 clients, 1.0 sampling rate, 20 communication rounds, 3 local epochs, micro-batch size 8, effective batch size 128, learning rate 3e-4, and LoRA with rank = 8, alpha = 16. We will clarify these details in the revised version for reproducibility.
>
> > **Response to Weakness 4 / Question 5**
>
> We agree that the credibility of metrics in Table 4 depends on a transparent evaluation protocol. So we actually used position swapping (A/B and B/A) and averaged results across 5 independent random samplings to get the results.
>
> Moreover, the emphasis on conciseness is not arbitrary, but captures a core symptom of the Rewrite Paradox: As shown in Figure 3, rewritten responses often become longer, less information-dense, and more stylistically biased. So evaluating redundancy and information purity directly targets the mechanism studied in this paper.
>
> In addition, thank you for your suggestion regarding the addition of manual assessment. Our manual evaluation of the win rate indeed confirms the same trend as Table 4, suggesting alignment with human judgment:
>
> | Method | Model Win Rate (%) | Human Win Rate (%) |
> | :--- | :---: | :---: |
> | FedAvg | 5.80 | 4.35 |
> | Distill Rect | 4.50 | 3.75 |
> | Dual Upload | 3.25 | 2.60 |
> | FedDPA | 10.35 | 9.85 |
> | FedSD (Ours) | 8.30 | 8.65 |
> | **FedSDR (Ours)** | **67.75** | **70.80** |
>
> We will strengthen the final version by describing the debiased evaluation protocol more clearly and adding human evaluation results as supplementary validation.
>
> ---
>
> [1] OpenFedLLM: Training Large Language Models on Decentralized Private Data via Federated Learning
>
> [2] Dual-Personalizing Adapter for Federated Foundation Models

---

> > ### Author Rebuttal · Reviewer_ofrB · 2026-04-01
> >
> > Some experiments have been done to address my questions.
> > but some questions are answered with simple explanation, which does not fully address the original questions.

---

> > > ### Author Response · Authors · 2026-04-01
> > >
> > > Dear Reviewer,
> > >
> > > Thank you for your prompt review of our rebuttal and for acknowledging the additional experiments. We are very eager to address your follow-up questions to ensure all your concerns are fully resolved. It appears the specific questions may not have been attached to your latest comment—could you please share them when you have a moment?
> > >
> > > Regarding your note that some questions were answered with a "simple explanation," we would be very grateful if you could point out which specific responses felt insufficient. Aside from our clarification on the hyperparameter settings where we deferred to established baseline configurations, we strived to support all of our responses to your questions and weaknesses with concrete empirical evidence. This included providing new cross-task gradient and loss transfer analyses, conducting additional experiments with the Qwen2.5-7B backbone on diverse domain benchmarks, and conducting new human evaluation validations.
> > >
> > > If our explanations regarding any of the newly provided empirical results lacked the necessary depth, please let us know so we can expand on them. We stand ready to provide further clarifications or run additional analyses as soon as you share your follow-up questions. Thank you again for your time reviewing our rebuttal!
> > >
> > > Best regards,
> > >
> > > The Authors

---

### Official Review · Reviewer_nqfT · 2026-03-13

**Soundness:** 3
**Presentation:** 2
**Significance:** 3
**Originality:** 2
**Overall Recommendation:** 4
**Confidence:** 3

**Summary:**

The paper addresses statistical heterogeneity in federated LLM fine-tuning by Federated Self-Distillation to project client data into a shared representation space. Federated SD smooths optimization but introduces factual hallucinations and redundancies. To resolve this conflict, the authors develop a dual-stream architecture that uses a local smoothing adapter and a globally aggregated rectification adapter anchored to raw data to maintain factual correctness.

**Compliance With Llm Reviewing Policy:**

Affirmed.

**Final Justification:**

All my questions have been resolved.

**Key Questions For Authors:**

- How sensitive are the results to the specific self-distillation procedure, including the rewrite prompt, decoding strategy, and the choice to perform rewriting only once before federated training?

- How the computational overhead of generating the distilled dataset and performing the alternating dual-stream optimization compares to standard federated fine-tuning methods?

- Whether the severity of the Rewrite Paradox varies depending on the innate capabilities of the chosen seed model?

**Limitations:**

Yes

**Strengths And Weaknesses:**

**Strengths**

- Federated fine-tuning of LLMs under severe non-IID data is a meaningful setting, and the paper starts from a data-centric perspective that is somewhat different from the usual optimizer or adapter-centric approach.

- Proposed “Rewrite Paradox” gives a concrete motivation for why naive self-distillation may help optimization while hurting faithfulness.


**Weaknesses**

- The paper argues that self-distillation aligns heterogeneous client data into a shared space and thereby reduces client drift, but the evidence is indirect. The t-SNE visualization do not really establish that data-level alignment drives improvement, as opposed to simpler effects such as regularization, label smoothing.

- Experimental scope is narrower than the framing. Main experiments on Databricks Dolly-15K and simulated non-IID splits is insuffcient to support the broad claims about federated LLM fine-tuning, unless the setting includes more diverse datasets, client partition strategies, model scales.

- The paper innovation should be framed more carefully, as self-distillation in federated learning, dual LoRA or adapter designs, and selective aggregation of low-rank components each already have close prior works [1-3].


[1] Adaptive Self-Distillation for Minimizing Client Drift in Heterogeneous Federated Learning

[2] FDLoRA: Personalized Federated Learning of Large Language Model via Dual LoRA Tuning

[3] Selective Aggregation for Low-Rank Adaptation in Federated Learning

---

> ### Author Rebuttal · Authors · 2026-03-31
>
> Thank you for the constructive feedback. Below we respond your questions (Q) and weaknesses (W) point by point.
>
> > **W1**
>
> To more directly prove self-distillation aligns heterogeneous data, we further added two analyses on five highly heterogeneous domain datasets (**FinGPT, MathInstruct, Code-Alpaca, Alpaca, and MedAlpaca**): **cross-task gradient cosine similarity** and **cross-task loss transfer**.
> We summarize for each source task the average scores over the other four target tasks. Here, **Grad. Sim.** denotes the cross-task gradient cosine similarity, and **Loss Trans.** denotes the average change in loss for other tasks before and after the model was trained on a certain source task. Below are the rate of change of these two matrics after the self-distillation process:
>
> | Source task | Grad. Sim. | Loss Trans. |
> |---|---:|---:|
> | FinGPT | +41.5 pp | -2.5 pp |
> | MathInstruct | +13.8 pp | -4.0 pp |
> | Code-Alpaca | +13.6 pp | -4.9 pp |
> | Alpaca | +10.8 pp | -5.1 pp |
> | MedAlpaca | +8.1 pp | -5.4 pp |
>
> All five source tasks show positive gains in gradient similarity and negative loss changes, meaning that after self-distillation, update directions across tasks become more aligned overall.  These analyses provide mechanism-level support for our interpretation: self-distillation **reshapes heterogeneous supervision into a more aligned and cooperative optimization space**.
>
> > **W2**
>
> To address the concern of experimental scope, we added experiments across both data distributions and model families. We use the new mixed domain dataset mentioned above, replaced the backbone with Qwen2.5-7B-Instruct and evaluate in each corresponding domain benchmarks. FedSDR still achieves consistent gains in this more heterogeneous setting:
>
> | Benchmark | Baseline | Ours | Delta |
> |---|---:|---:|---:|
> | MMLU-Med | 0.6246 | 0.6957 | +0.0711 |
> | FPB | 0.9073 | 0.9227 | +0.0154 |
> | HumanEval | 0.5244 | 0.6280 | +0.1037 |
> | GSM8K | 0.6679 | 0.7832 | +0.1153 |
>
> > **W3**
>
> Actually, all these prior works have already been discussed in our submission, and related directions such as dual-LoRA and selective aggregation (like FedDPA) are also included in our baselines.
> Our contribution is not to claim self-distillation, dual LoRA, or selective aggregation as standalone novelties. Rather, our key point is that federated LLM personalization should be viewed not only from the model-design perspective, but also from the data-distribution perspective. In particular, we highlight the role of intrinsic supervision mismatch across clients and address it through a data-centric self-distillation framework.
> From this perspective, to the best of our knowledge, we are the first to extend self-distillation to federated LLM personalization by using the model’s own understanding space to normalize heterogeneous supervision. **Empirically, this data-centric view improves not only classical FL methods with a single LoRA, but also stronger LLM+FL frameworks such as dual-LoRA and selective aggregation, suggesting that it is complementary to prior model-centric work rather than competing with it.**
>
> > **Q1**
>
> In this paper, we intentionally follow prior self-distillation practice [1] by using a basic rewrite prompt with greedy decoding to avoid attributing the gains to a carefully tuned distillation recipe. We adopt the one-shot rewriting strategy for practical efficiency, which keeps the method realistic under federated resource constraints. Our goal is not to search for the optimal rewriting recipe, but to verify whether self-distillation works as a practical data-centric mechanism. Even under this simple setup, the method still shows stable heterogeneity-mitigation effects.
>
> > **Q2**
>
> It is important to note that our method is mainly applied to enhance the existing classic FL algorithms. Therefore, within each algorithm, only the one-time local preprocessing accounts for 1% to 5% extra computation overhead. Moreover, in the dual-stream training stage, just as stated in our baseline work that serves as the foundation for this study [2], the LoRA architecture allows the local LoRA and global LoRA to be trained in parallel. As a result, the additional computation overhead from this part is only about 2%.
>
> > **Q3**
>
> The severity of the Rewrite Paradox does depend on the capability of the seed model, since it arises from the model re-amplifying its own inductive biases during rewriting.
> We therefore view the Rewrite Paradox not as a backbone-specific artifact, but as a more general phenomenon whose dominant form varies with model capability. This is exactly why FedSDR does not assume the seed model is fully reliable. Accordingly, LoRA-R stays anchored to raw-data supervision, while LoRA-S is used only locally to absorb both the smoothing benefit, without participating in global aggregation.
>
> ---
>
> [1] Self-Distillation Bridges Distribution Gap in Language Model Fine-Tuning
>
> [2] Dual-Personalizing Adapter for Federated Foundation Models

---

> > ### Author Rebuttal · Reviewer_nqfT · 2026-04-01
> >
> > Most of my questions have been addressed. Improving the related work discussion and adding the experiment during rebuttal would further strengthen the paper. Since my initial assessment is positive I will maintain the score.

---

> > > ### Author Response · Authors · 2026-04-04
> > >
> > > Dear Reviewer,
> > >
> > > Thank you very much for your thoughtful feedback throughout this review process. We are thrilled to hear that our rebuttal addressed your concerns and that you maintain a positive assessment of our work.
> > >
> > > We completely agree with your suggestion regarding the manuscript. In the revised version, we will ensure that the new cross-task gradient/loss transfer experiments, the additional results on the Qwen backbone across diverse domains, and the clarified discussion on related works are prominently integrated to further strengthen the paper.
> > >
> > > Thank you again for your time and support of our research!
> > >
> > > Best regards,
> > >
> > > The Authors

---

### Decision · Program_Chairs · 2026-04-30

**Decision:**

Accept (regular)

**Comment:**

This paper received borderline reviews (3, 4, 4, 4). Reviewers are generally satisfied with the rebuttal, while have remaining concerns. More specifically, Reviewer nqfT requested the authors to keep improving the related work discussion and adding the experiment during rebuttal to the paper draft. Reviewers ofrB, 1usw and rKhU mentioned their concerns are partially resolved, but did not provide further actionable requests. Overall, this is a borderline paper that shows some values.